# Incidence and predictors of postpartum depression among postpartum mothers in Kuala Lumpur, Malaysia: A cross-sectional study

Mohd Izzuddin Hairol [1]⊙*, Sha'ari Ahmad[1]⊙, Sharanjeet Sharanjeet-Kaur[2]⊙, Lei Hum Wee[1]⊙, Fauziah Abdullah[3]⊙, Mahadir Ahmad [1]⊙

1 Centre for Community Health Studies (ReaCH), Faculty of Health Sciences, Universiti Kebangsaan Malaysia, Kuala Lumpur, Malaysia, 2 Centre for Rehabilitation and Special Needs Studies, Faculty of Health Sciences, Universiti Kebangsaan Malaysia, Kuala Lumpur, Malaysia, 3 Jabatan Kesihatan Wilayah Persekutuan Kuala Lumpur & Putrajaya (The Federal Territory of Kuala Lumpur & Putrajaya Health Department), Kuala Lumpur, Malaysia

⊙ These authors contributed equally to this work.

* izzuddin.hairol@ukm.edu.my

**Data Availability Statement:** All relevant data are within the paper and its Supporting Information files.

## Abstract

Postpartum depression (PPD) is one of the mental health complications that may arise following childbirth. This cross-sectional study explores the association between socioeconomic factors and PPD literacy with PPD incidence in 350 participants (mean age: 30.58 ±4.72 years) at one to six months postpartum, who attended the Kuala Lumpur Health Clinic from May to October 2020. PPD incidence and literacy were assessed using the validated Malay versions of the Edinburgh Postpartum Depression Scale (EPDS) and the Postpartum Depression Literacy Scale (PoDLiS), respectively. The participants' socioeconomic characteristics were collected using a self-administered questionnaire. Chi-square tests were performed to determine the association between these factors and PPD incidence. Binary logistic regression models were used to determine the odds ratios (OR). The incidence of postpartum depressive symptoms was 14.29%. Those with low household income were twice likely to have PPD symptoms (OR:2.58, 95% CI:1.23–5.19; $p = 0.01$) than those with higher incomes. Unemployment (i.e., participants who were housewives/homemakers) was associated with higher PPD incidence ($X^2_{(2, 350)} = 6.97$, $p = 0.03$), but it was not a significant PPD predictor. In conclusion, PPD incidence in the sample of Kuala Lumpur postpartum mothers is significantly associated with low household income. Other socioeconomic characteristics, including PPD literacy, were not significant predictors of PPD incidence.

## Introduction

Postpartum depression (PPD) is one of the mental health complications that may arise following childbirth. The Diagnostic and Statistical Manual of Mental Disorders Fifth Edition (DSM-5) [1] defines postpartum depression as a depressive episode with moderate to severe

**Funding:** Author MIH received funding from the Tun Fatimah Hashim Women's Leadership Centre, Universiti Kebangsaan Malaysia (https://www.ukm.my/pkwtfh/ms/), and the Ministry of Women, Family, and Community Development, Malaysia (https://www.kpwkm.gov.my/), grant number PKW-2019-004. The funders had no role in the study design; data collection, analysis, and interpretation; report writing; and in the decision to submit the article for publication.

**Competing interests:** The authors have declared that no competing interests exist.

severity that begins four weeks after delivery and can last up to 12 months after childbirth [2]. The reported worldwide prevalence of postpartum depression varied widely but with an overall prevalence of 17.7% [3]. PPD leads to adverse consequences for the mother, the infant's development, and the family environment [4, 5].

In Malaysia, the range of postpartum depression prevalence is between 3.9% and 22.8% [6–9]. The reported prevalence varies by state in the country. In the north-eastern Kelantan region, the prevalence of postpartum depression was between 9.8% [10] to 22.8% [8]. In the eastern Sabah region, the prevalence was 14.3% [7], while the capital Kuala Lumpur, had a prevalence of 3.9% [6].

There are various risk factors for postpartum depression. In a review, a history of depression, stressful life events, low social support, antenatal anxiety, preference of infant's gender, and low income were reported to be the pertinent risk factors [11]. Specifically for Malaysian mothers, there were significantly higher risks for PPD among those who had depression during pregnancy [7, 8, 12] and with constant worries about their child [7]. Marital problems and low household income groups were also associated with PPD incidence [7, 10]. Malaysian women having emergency delivery had twice the risk of developing PPD [9]. PPD incidence tends to be higher among mothers who lived with their extended family [6]. Implementation of traditional postpartum practice was associated with a higher incidence of PPD among those living in relatively rural areas [8] but not among those living in urban Kuala Lumpur [6]. Unlike their counterparts in several Asian countries such as India [13] and China [14], PPD incidence in Malaysian mothers was not significantly associated with an unplanned pregnancy and the infant's gender [7, 8, 10]. The most recent on PPD incidence and risk factors were reported for Malaysian mothers residing in relatively rural areas of the country. On the other hand, the PPD incidence in the capital city, Kuala Lumpur, were reported about 20 years ago [6, 9].

The consequences of depressive symptoms of postpartum depression could be minimized with effective treatments [15, 16]. However, women with postpartum depression must have knowledge about PPD signs and symptoms and its treatment possibilities before professional help can be sought. If postpartum women are encouraged and equipped with adequate knowledge on postpartum depression symptoms, risk factors, causes, and treatments, they are more likely to seek support, treatment, and professional help [17–19]. Studies related to PPD knowledge and awareness among Malaysian mothers are limited, where a study reported 51.9% of their Malaysian postpartum respondents had heard of women getting sad after childbirth [6]. It is not known if PPD awareness and literacy are associated with PPD incidence among Malaysian mothers.

This study's main objective was to determine the predictors of postpartum depression incidence in a sample of postpartum mothers in Kuala Lumpur, Malaysia. These predictors include socioeconomic characteristics and the level of PPD literacy and awareness. The secondary objective was to determine the latest postpartum depression incidence in the same sample of participants.

## Materials and method

### Participants

This cross-sectional study involved postpartum women attending the Kuala Lumpur Health Clinic for their scheduled infant's immunization and routine examination. The clinic was chosen based on the Medical Research Ethics Committee (MREC), Ministry of Health's approval. Women who attended the clinic between one to six months postpartum were invited to participate in this study. All participants were selected via convenient sampling. The inclusion

criteria included age 18 years and older and able to speak, read, and write in Malay or English. Those who were deemed unwell to participate, as advised by health professionals, were excluded. Sample size, $n$, is calculated with the formula

$$n = z^2 pq/d^2$$

where $z = 1.96$, $p = 22.8\%$ which is the published prevalence of postpartum depression in Malaysia [8], $q = 1—p$ and $d = 0.05$. Taking possible dropouts into account, the final sample size for this study was 350.

## Instruments

The Edinburgh Postpartum Depression Scale (EPDS) validated Malay version [20, 21] was used to measure postpartum depression. It is a 10-item self-administered questionnaire where each item was rated on a 4-point scale. The total score ranged between 0 and 30. In the verified Malay version of the EPDS, a score of 11.5 was the optimum cut-off point for 72.7% sensitivity, 95% specificity, and a positive predictive value of 80% [21]. In our study, participants with an EPDS score of ≥12 were categorized as having postpartum depressive symptoms.

The Postpartum Depression Literacy Scale (PoDLiS) is a 31-item self-administered questionnaire designed to assess knowledge of postpartum depression [22]. These items were categorized into seven attributes related to postpartum depression literacy, which were (1) the ability to recognize postpartum depression; (2) knowledge of risk factors and causes; (3) knowledge and beliefs in self-care activities; (4) knowledge about professional help available; (5) belief about professional help available; (6) attitudes which facilitation recognition of postpartum depression and appropriate help-seeking; and (7) knowledge of how to seek information related to postpartum depression. Each item was rated on a 5-point scale. Each attribute's score was calculated by adding the raw scores for all related items and dividing it with each attribute's number of items. The total score was calculated by dividing the total raw score by the total number of items.

The PoDLis was first translated from English into the Malay language. First, two bilingual translators translated it from English into Malay using the back-translation method. Then, two bilingual specialists from the linguistic department at a regional university in Malaysia, who were unexposed to the original English version, were engaged in the backward translation. The Malay version was compared to the initial English version, and relevant amendments were made. The final Malay version comprised of 31 items identical in their senses and background to the English version. The original and the back-translated versions were compared to determine its face and content validity. Six experts were asked to determine the relevance of the items using a 4-point Likert scale to calculate the Content Validity Index (CVI). The average CVI score across all items (S-CVI/average) was calculated and found to be 0.84, which was satisfactory [23]. The translated version was also tested on 535 female participants. The Malay PoDLiS had good reliability indicated by internal consistency, with Cronbach's alpha coefficient of 0.73.

## Procedures

All potential participants were briefed about the research objectives, and written consent was obtained from the participants. The participants were then provided with a questionnaire that included information on demographics and socioeconomic status. They were also provided with the Malay version of the Edinburgh Postpartum Depression Scale (EPDS) and the Postpartum Depression Literacy Scale (PoDLiS). A graduate student in Psychology (author SA)

circulated the questionnaires to the participants. The participants took approximately 20 minutes to complete all questionnaires. Author SA collected all questionnaires on the same day.

Demographic and socioeconomic status, including participants' ethnicity, postpartum period, the number of children in the family, monthly household income, maternal education status, maternal employment status, living arrangements, and birth location, was obtained using a self-administered questionnaire. Factors such as the infant's gender and unplanned pregnancy were not included, as earlier reports in similar populations found that these factors had no significant association with PPD incidence [7, 8, 12].

## Statistical analyses

The characteristics of the participants were analyzed with descriptive analysis. Two-tailed independent-samples t-tests were performed to compare the differences in age and PoDLiS score between participants with and without PPD depressive symptoms (EPDS score $\geq 12$ and $< 12$, respectively). Chi-square tests were performed to determine the association between socioeconomic predictors and PPD literacy with the incidence of postpartum depression. Any predictors that produced a p-value less than 0.05 in the chi-square tests were included in the binary logistic regression model. Odds ratios (OR) were determined using logistic regression analyses after adjusting for confounding variables. ORs with 95% confidence intervals (CI) were used to assess the associations' magnitude. The first regression model was unadjusted, while the second regression model was adjusted by the participants' age, monthly household income, and employment status. The significance level, $\alpha$, was set at 0.05. All analyses were conducted using the IBM SPSS Statistical Package Version 22 (IBM, Armonk, NY, USA).

## Ethics consideration

Ethics approval for this study was obtained from the Faculty of Health Sciences Research Ethics Committee, Universiti Kebangsaan Malaysia (UKM) (approval code: UKM PPI /111 /8 /JEP2020-316) and the Medical Research Ethics Committee (MREC), Ministry Health, Malaysia (approval code: NMRR-19-4013-52121 (IIR)). The conduct of this study adhered to the principles of the Declaration of Helsinki and the Malaysia Good Clinical Practice Guidelines.

## Results

In total, 350 participants completed the study. The participants' mean age was 30.58 ± 4.72 years, while the infants' mean age was 2.08 ± 1.53 months. Table 1 summarizes the participants' demographic characteristics.

Fifty participants had an EPDS score of $\geq 12$. Therefore, the incidence of postpartum depressive symptoms was 14.29%. The mean age of participants with PPD was lower (29.02 ±5.44 years) compared with those without PPD (30.84±4.54 years). This difference was statistically significant [$t_{(348)} = 2.55$, $p = 0.03$]. Participants with PPD had a lower PoDLiS score (3.68 ±0.38) compared to those without PPD (3.78±0.37). However, this difference was not significant [$t_{(348)} = 1.90$, $p = 0.06$].

The incidence of PPD was significantly higher for participants from the low-income group (27.27%) compared to those from the middle- and high-income groups (8.33%) [$X^2_{(2, 350)} = 14.49$, $p = 0.01$]. Mothers who were housewives or homemakers also had a significantly higher incidence of postpartum depressive symptoms (30.16%) compared to those who were employed (13.37%) or self-employed (11.43%) [$X^2_{(2, 350)} = 6.97$, $p = 0.03$]. Depressive symptom incidence was highest in the first month of postpartum (16.8%) compared to more extended postpartum periods (around 11%); however, this difference was not statistically significant ($p = 0.33$). There was no significant difference in the incidence of postpartum depression

**Table 1. Socioeconomic characteristics of the participants (N = 350) in this study.**

| Socioeconomic characteristics | N | % |
|---|---|---|
| Ethnicity | | |
| • Malay | 306 | 87.43 |
| • Chinese | 23 | 6.57 |
| • Indian | 13 | 3.71 |
| • Others | 8 | 2.29 |
| Postpartum period (months) | | |
| • 1 | 197 | 56.20 |
| • 2–3 | 110 | 31.43 |
| • 4–6 | 63 | 18.00 |
| Number of children | | |
| • 1 | 125 | 35.70 |
| • 2 | 104 | 29.70 |
| • 3 and above | 121 | 34.60 |
| Monthly household income | | |
| • Low (<MYR4360) | 168 | 48.00 |
| • Middle (MYR4360 to MYR9619) | 163 | 46.60 |
| • High (≥MYR9619) | 19 | 5.40 |
| Maternal education status | | |
| • Low (Up to secondary school) | 105 | 30.00 |
| • Medium (Up to pre-university/diploma) | 130 | 37.10 |
| • High (Completed tertiary education) | 115 | 32.90 |
| Maternal employment status | | |
| • Employed | 229 | 65.40 |
| • Self-employed | 39 | 11.10 |
| • Housewife/homemaker | 82 | 23.40 |
| Living arrangements | | |
| • Own house | 257 | 73.43 |
| • With participant's extended family | 38 | 10.86 |
| • With spouse's extended family | 55 | 15.71 |
| Birth location | | |
| • Government hospital | 313 | 89.40 |
| • Private hospital | 36 | 10.30 |
| • Own house | 1 | 0.30 |

symptoms between ethnicity, the number of children in the family, maternal education status, living arrangements, and birth location (all $p > 0.05$). Table 2 summarizes the incidence of PPD based on the participants' socioeconomic characteristics and PPD literacy.

Table 3 shows the odds ratio (OR) of the binary logistic regression models, where the included predictors were monthly household income, maternal employment status, mean PoDLis score, and mother's age. The high- and middle-income groups were combined for household income due to a small number of participants in the high-income group. When monthly household income were re-categorized into two groups (low: less than MYR4360) and middle and high (equal or more than MYR4360) (USD1 = MYR4.11, as of 21 September 2020), the PPD incidence for the low income group was significantly higher than the middle-high income group [$X^2_{(2, 350)} = 13.46$, $p < 0.001$].

The adjusted regression model reveals that participants from low monthly household income were 2.58 times more likely to have postpartum depressive symptoms (odds ratio:

**Table 2. Postpartum depression (PPD) incidence based on the study participants' socioeconomic characteristics & PPD literacy.**

| Participants' characteristics | PPD incidence | | | | $X^2$ or $t$ | $p$-value |
|---|---|---|---|---|---|---|
| | Yes (n = 50) | | No (n = 300) | | | |
| | n | % | n | % | | |
| Mean age | 29.02±5.44 | | 30.84±4.54 | | 2.55 | 0.03 |
| Mean PoDLiS score | 3.68±0.38 | | 3.78±0.37 | | 1.90 | 0.06 |
| Ethnicity | | | | | 4.89 | 0.18 |
| • Malay | 40 | 15.04 | 266 | 84.96 | | |
| • Chinese | 4 | 21.05 | 19 | 78.95 | | |
| • Indian | 3 | 30.00 | 10 | 70.00 | | |
| • Others | 3 | 60.00 | 5 | 40.00 | | |
| Postpartum period (months) | | | | | 2.24 | 0.33 |
| • 1 | 33 | 18.80 | 164 | 83.20 | | |
| • 2–3 | 10 | 11.10 | 80 | 88.90 | | |
| • 4–6 | 7 | 11.10 | 56 | 88.90 | | |
| Number of children in the family | | | | | 5.54 | 0.06 |
| • 1 | 25 | 20.00 | 100 | 80.00 | | |
| • 2 | 10 | 9.60 | 94 | 90.40 | | |
| • 3 and above | 15 | 12.40 | 106 | 87.60 | | |
| Monthly household income | | | | | 14.49 | 0.01 |
| • Low | 36 | 27.27 | 132 | 78.60 | | |
| • Middle | 14 | 8.60 | 149 | 91.40 | | |
| • High | 0 | 0.00 | 19 | 100.00 | | |
| Maternal education status | | | | | 0.97 | 0.61 |
| • Low | 18 | 17.10 | 87 | 82.90 | | |
| • Medium | 17 | 13.10 | 113 | 86.90 | | |
| • High | 15 | 13.00 | 100 | 87.00 | | |
| Maternal employment status | | | | | 6.97 | 0.03 |
| • Employed | 27 | 13.37 | 202 | 86.63 | | |
| • Self-employed | 4 | 11.43 | 35 | 88.57 | | |
| • Housewife/homemaker | 19 | 30.16 | 63 | 69.84 | | |
| Living arrangements | | | | | 0.35 | 0.55 |
| • Own house | 35 | 15.77 | 222 | 84.23 | | |
| • With participant's extended family | 6 | 18.75 | 32 | 81.25 | | |
| • With husband's extended family | 9 | 19.57 | 46 | 80.43 | | |
| Birth location | | | | | 2.69 | 0.26 |
| • Government hospital | 48 | 15.30 | 265 | 84.70 | | |
| • Private hospital | 2 | 5.60 | 34 | 94.40 | | |
| • Own house | 0 | 0.00 | 1 | 100 | | |

2.58, 95% CI: 1.23–5.19; $p$ = 0.01) compared to those with middle- to high- monthly household incomes. Mother's age and maternal employment status did not contribute significantly to the incidence of postpartum depressive symptoms for our study participants.

## Discussion

In this study, postpartum depression incidence for Malaysian mothers attending the Kuala Lumpur Health Clinic within the first six months postpartum was 14.29%. This figure is much higher than that reported by a study involving mothers in Kuala Lumpur conducted 20 years ago, which was 0.60% [6]. In Grace et al.'s study, the authors might have found a much lower

Table 3. Binary logistic regression analysis.

| Predictors | Unadjusted model | | Adjusted model | |
|---|---|---|---|---|
| | Odds ratio [95% CI] | P-value | Odds ratio [95% CI] | p-value |
| Participant's age | 0.92 [0.86, 0.98] | 0.01* | 0.94 [0.88–1.00] | 0.07 |
| Monthly household income | | <0.001** | | 0.01* |
| • <MYR4360 | 3.27 [1.70, 6.31] | | 2.58 [1.23, 5.19] | |
| • ≥MYR4360[b] | 1 | | 1 | |
| Maternal employment status | | | | |
| • Employed | 0.44 [0.23, 0.85] | 0.01* | 0.64 [0.32, 1.28] | 0.21 |
| • Self-employed | 0.38 [0.12, 1.20] | 0.10 | 0.51 [0.16, 1.66] | 0.27 |
| • Housewife/homemaker[c] | 1 | | 1 | |

[a]The reference category for ethnicity is the Other ethnic groups.

[b]The reference category for monthly household income is the middle- to high-income group (≥MYR4360).

[c]The reference category for maternal employment status is unemployed.

*significant at p<0.05

**significant at p<0.01.

PPD incidence as they did their Malay translation of the EPDS and used a cut-off score of ≥13 to determine the case for postpartum depression. On the other hand, our study used the validated Malay version of the EPDS with a lower cut-off score of 11.5 [21], contributing to the differences in our findings.

The PPD incidence reported in our study compares well to that reported in the state Sabah, East Malaysia (using validated Malay version of EPDS: 14.3%) [7], and in mothers who underwent emergency delivery (using English version of EPDS: 16.7%) [9]. However, it is lower than that reported in the state of Kelantan (22.8%) [8]. The different PPD incidence between different regions in Malaysia could be related to differences in household incomes in these areas and how these authors defined them. The higher incidence in Kelantan could be due to the state having the lowest Gross Domestic Product (GDP) per capita in Malaysia [24]. Indeed, it has been widely reported that low household income increases the risk of postpartum depression incidence in developing countries [17, 25, 26].

Interestingly, although the state of Sabah had GDP per capita five times lower than Kuala Lumpur, the reported PPD incidence was similar to ours—but the household income was not a significant risk factor for PPD [7]. That study's findings may be attributed to how their participants' household incomes were defined. We used a monthly household income of less than MYR4360 to define low income, as the figure is the ceiling income bracket for the bottom 40% of household earners in Malaysia. In contrast, a previous study used MYR3000 as the upper threshold for their low-income group [7]. The difference in income categorization could explain why the incidence of postpartum depression in this study is similar to theirs, despite Sabah having a much lower GDP per capita than Kuala Lumpur. A higher PPD incidence might be expected if the study had used a similar ceiling to define their low-income group.

We also found a higher incidence of postpartum depression among postpartum mothers who were unemployed (i.e., housewives or homemakers) than those in employment. Our findings agree with those of other studies on Malaysian populations [7, 10, 12, 27]. Although the incidence was highest among postpartum mothers who were unemployed, we did not find employment status a significant predictor of postpartum depression. Conflicting findings have been reported as to whether employment status is a definite predictor for postpartum depression symptoms. For example, employed mothers were more likely to have fewer postpartum

depression symptoms [28] while the opposite have also been reported [29]. It is likely that multiple work-related factors, such as workload, organizational support, and relationship with colleagues, also play a role. In other words, these factors could help a mother ease her transition back to work or exacerbate the symptoms of PPD. Therefore, future studies must consider these work-related factors that may contribute to the incidence of postpartum depression, rather than just the mother's employment status.

Besides, we did not find a significant association between maternal education status and the incidence of postpartum depression. These findings could be explained by the relatively higher household income, multiple job opportunities, and vast sources of information that were accessible by Kuala Lumpur residents. These better socioeconomic opportunities might have offset the effects of maternal education status on postpartum depression incidence. Low education status is a significant determinant for postpartum depression in countries such as Turkey [30], Vietnam [31], and Japan [32]. However, the definition of 'low education' differs between studies, such as completion of primary schooling or less [30] or ≤12 years of education [32]. We defined it as completion of secondary schooling at the age of 17. This difference in the definition could explain why we did not find a significant association between education levels and PPD incidence in our participants. Although a study reported that maternal education is a significant risk factor for PPD for Japanese mothers, their adjusted regression model showed that the association was not for those with up to 16 years of education [32]. Their findings are similar to ours when the number of years of maternal education were similar.

Maternal age is a known risk factor for developing postpartum depression. In our unadjusted logistic regression model, younger mothers had significantly higher risks for having postpartum depression symptoms. However, in the adjusted logistic regression model, the association between the mother's age and the postpartum depression symptoms was no longer significant. There are reports that younger maternal age is a positive risk factor for postpartum depression [33, 34], where the risk is highest for maternal age younger than 25 years old [34]. However, several studies have found no such association [35, 36]. In our study, participants younger than 25 years old made up only 6.29% of the total sample size, which could partly explain why no significant association was found between maternal age and postpartum depression incidence in our adjusted regression model.

Although we did not find PPD literacy to be a significant predictor for PPD incidence, participants with PPD in our study had lower scores for PPD literacy and awareness than those without PPD. Our results suggest that our sample of postpartum mothers were aware of PPD, regardless of whether they had PPD symptoms or not. However, awareness of PPD may not necessarily translate to actions or attitudes that facilitate recognition of postpartum depression and appropriate help-seeking. It has been reported that the acceptance of psychiatric diagnosis and treatment is relatively low [37] among Malaysians, as the term psychiatric illness and seeking professional help have a very negative connotation and stigma in the community [38]. Indeed, the stigma and negative opinions act as barriers that prevent women with PPD from seeking professional help [39, 40]. Therefore, public education campaigns must be broad-reaching to target all expecting and postpartum women. It could be made more efficient by targeting segments in the population who are more likely to have a higher PPD incidence and a lower knowledge of PPD [41]. There are public awareness campaigns in Malaysia using infographics and guidelines published by the Ministry of Health [42, 43] and newspaper articles [42, 44]. Their effectiveness in increasing knowledge and changing attitudes on PPD could be evaluated in future studies.

This study has limitations that should be considered when interpreting the findings. First, the PPD literacy level between participants with and without PPD was differentiated by comparing each group's mean scores, as measured with the PoDLiS. A cut-off point for the scores

may be more practical to categorize a participant's PPD literacy and awareness levels, which could be the direction of a future study. Second, we did not include some factors such as antenatal depression history and satisfaction with marital relationships as the predictors for the incidence of postpartum depression symptoms. Third, although the PPD incidence found for mothers in the capital Kuala Lumpur was similar to that reported from other parts of Malaysia, the predictors associated with PPD incidence that we reported may apply to postpartum mothers residing in urbanized areas only.

## Conclusions

The incidence of postpartum depression measured among mothers in Kuala Lumpur using the Malay version of the EPDS is 14.29%. PPD incidence is significantly associated with low household income. Those with PPD are also statistically younger than those without PPD symptoms. Other socioeconomic characteristics, including PPD literacy level, were not significant predictors of PPD incidence. Therefore, public education campaigns could be made more efficient by targeting those who are more likely to develop PPD symptoms during antenatal and early postpartum periods.

## Supporting information

**S1 Table. Socioeconomic characteristics of the participants (N = 350) in this study.**
(DOCX)

**S2 Table. Postpartum depression (PPD) incidence based on the study participants' socioeconomic characteristics & PPD literacy.**
(DOCX)

**S3 Table. Binary logistic regression analysis.** [a]The reference category for ethnicity is the Other ethnic groups. [b]The reference category for monthly household income is the middle- to high-income group ($\geq$MYR4360). [c]The reference category for maternal employment status is unemployed. *significant at $p<0.05$ **significant at $p<0.01$.
(DOCX)

**S1 Data.**
(XLSX)

## Acknowledgments

We would like to thank the Director-General of Health Malaysia for his permission to publish this article, and the Director of the Kuala Lumpur and Putrajaya Health Department and the Chief and staff of the Kuala Lumpur Health Clinic for their assistance.

## Author Contributions

**Conceptualization:** Mohd Izzuddin Hairol, Sharanjeet Sharanjeet-Kaur, Lei Hum Wee, Fauziah Abdullah, Mahadir Ahmad.

**Data curation:** Mohd Izzuddin Hairol, Sha'ari Ahmad.

**Formal analysis:** Mohd Izzuddin Hairol, Sha'ari Ahmad.

**Funding acquisition:** Mohd Izzuddin Hairol, Sharanjeet Sharanjeet-Kaur, Lei Hum Wee, Fauziah Abdullah, Mahadir Ahmad.

**Investigation:** Sha'ari Ahmad.

**Methodology:** Mohd Izzuddin Hairol, Sha'ari Ahmad, Sharanjeet Sharanjeet-Kaur, Lei Hum Wee, Fauziah Abdullah, Mahadir Ahmad.

**Project administration:** Mohd Izzuddin Hairol.

**Resources:** Mohd Izzuddin Hairol, Sha'ari Ahmad, Fauziah Abdullah, Mahadir Ahmad.

**Software:** Mohd Izzuddin Hairol, Sha'ari Ahmad.

**Supervision:** Mohd Izzuddin Hairol.

**Validation:** Mohd Izzuddin Hairol, Sha'ari Ahmad, Sharanjeet Sharanjeet-Kaur, Lei Hum Wee, Fauziah Abdullah, Mahadir Ahmad.

**Visualization:** Mohd Izzuddin Hairol.

**Writing – original draft:** Mohd Izzuddin Hairol.

**Writing – review & editing:** Mohd Izzuddin Hairol, Sha'ari Ahmad, Sharanjeet Sharanjeet-Kaur, Lei Hum Wee, Fauziah Abdullah, Mahadir Ahmad.

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
