## [Decision Letter · Decision Letter 0]

8 Oct 2021

PONE-D-21-08937

Incidence and predictors of postpartum depression among postpartum mothers in Kuala Lumpur, Malaysia: a cross-sectional study

PLOS ONE

Dear Dr. Hairol,

Thank you for submitting your manuscript to PLOS ONE. After careful consideration, we feel that it has merit but does not fully meet PLOS ONE’s publication criteria as it currently stands. Therefore, we invite you to submit a revised version of the manuscript that addresses the points raised during the review process.

We look forward to receiving your revised manuscript.

Kind regards,

Hanani Tabana, Ph.D

Academic Editor

PLOS ONE

Journal Requirements:

3. For more information on PLOS ONE's expectations for statistical reporting, please see https://journals.plos.org/plosone/s/submission-guidelines.#loc-statistical-reporting. Please update your Methods and Results sections accordingly.

Reviewers' comments:

Reviewer's Responses to Questions

**Comments to the Author**

1. Is the manuscript technically sound, and do the data support the conclusions?

Reviewer #1: Yes

Reviewer #2: Yes

Reviewer #3: Yes

2. Has the statistical analysis been performed appropriately and rigorously? 

Reviewer #1: Yes

Reviewer #2: Yes

Reviewer #3: Yes

3. Have the authors made all data underlying the findings in their manuscript fully available?

Reviewer #1: Yes

Reviewer #2: Yes

Reviewer #3: Yes

4. Is the manuscript presented in an intelligible fashion and written in standard English?

Reviewer #1: Yes

Reviewer #2: Yes

Reviewer #3: Yes

5. Review Comments to the Author

Reviewer #1: Manuscript is easy to read with minor correction needed.

Abstract: to remove the bottom 40% group as it appears out of context. Line 172- to explain who is unemployed mother as in the table 1, the result only shows housewife/homemaker. Line 213, to remove the closing bracket, line 215 to complete the sentence (instead of using the bracket) ...with a lower for cut-off point of 11.5,and reference of Azidah should be numbered -not APA style. To rephrase line 216-219 as the sentences are hanging.

Reviewer #2: Thank you for giving me a chance to review your manuscript. Please clarify abstract result and keyword, in method please include the CVI result from your back translation to Malay. Result: Regression analysis only included variables that had significant association to PPD

Reviewer #3: The manuscript is well written and logical flow. The research methods used is appropriate. However, the knowledge gained from this research is not new in the filed of PDD. The selected factors are so simple and are already identified from the literature.

6. PLOS authors have the option to publish the peer review history of their article (what does this mean?). If published, this will include your full peer review and any attached files.

Reviewer #1: No

Reviewer #2: **Yes: **Irma Nurbaeti

Reviewer #3: No

---

## [Author Response · Author response to Decision Letter 0]

25 Oct 2021

Dr. Hanani Tabana

Academic Editor

PLOS ONE

Manuscript ID: PONE-D-21-08937

Title: Incidence and predictors of postpartum depression among postpartum mothers in Kuala Lumpur, Malaysia: a cross-sectional study

Dear Dr Tabana, 

Thank you for allowing us the opportunity to address the comments made by the Editorial Team and the three reviewers. We are pleased that the manuscript is found to be well-written. All comments have been addressed in the revised version of the manuscript and are listed below. 

Editor’s comments/journal requirements:

1. We can now confirm that the manuscript has met PLOS ONE’s style requirements.

2. The references have been reviewed, and we can confirm that they are complete and correct. 

3. The statistical reporting for Methods and Results has been updated according to PLOS ONE’s expectations. The text in Statistical Analysis (lines 140-155) has been modified accordingly. 

4. All data have been made available with the submission of the revised manuscript. 

5. Captions on Supporting Information have been included an the end of the manuscript. 

Reviewers’ comments

Reviewer #1 

1. Reviewer #1 commented that the manuscript was easy to read, and we thank them for that comment. 

2. Abstract: to remove the bottom 40% group as it appears out of context. 

Our response: 

The phrase ‘the bottom 40%’ has been removed in the Abstract as suggested. 

3. Line 172- to explain who is unemployed mother as in the table 1, the result only shows housewife/homemaker. 

Our response:

In line 180, the phrase ‘unemployed mothers’ has been replaced with ‘mothers who were housewives or homemakers.’ Now the use of the phrase is consistent between the text and Table 1. 

4. Line 213, to remove the closing bracket, 

Our response:

The closing bracket (now in line 222) has been removed. 

5. line 215 to complete the sentence (instead of using the bracket) ...with a lower for cut-off point of 11.5,

Our response:

In line 223, the sentence has been completed instead of using brackets. The sentence now reads, “..our study used the validated Malay version of the EDPS with a lower cut-off score of 11.5.”

6. and reference of Azidah should be numbered -not APA style. 

Our response:

The reference of Azidah et al. (2004) in line 224 has been numbered according to Plos One’s citation format. 

7. To rephrase line 216-219 as the sentences are hanging.

Our response: 

Lines 226-228 have been rephrased. The original sentence has been split into two to allow a better reading flow. It now reads: 

“The PPD incidence reported in our study compares well to that reported in the state Sabah, East Malaysia (using validated Malay version of EDPS: 14.3%) [7], and in mothers who underwent emergency delivery (using English version of EDPS: 16.7%) [9]. However, it is lower than that reported in the state of Kelantan (22.8%) [8].

Reviewer #2: 

We would like to thank Reviewer 2 for reviewing the manuscript. 

1. Please clarify abstract result and keyword, in method please include the CVI result from your back translation to Malay. 

Our response:

In the Abstract (line 8), we have added the phrase “validated” to describe the questionnaires used in the study.

We have revised the keywords for our study. They are now: postpartum depression incidence; postpartum depression literacy and awareness; Edinburgh Postpartum Depression Scale; Postpartum Depression Literacy scale.

We have also included the CVI results for the back-translated PoDLis questionnaire. From lines 118 to 124, it now reads:

“The original and the back-translated versions were compared to determine its face and content validity. Six experts were asked to determine the relevance of the items using a 4-point Likert scale to calculate the Content Validity Index (CVI). The average CVI score across all items (S-CVI/average) was calculated and found to be 0.84, which was satisfactory [23]. The translated version was also tested on 535 female participants. The Malay PoDLiS had good reliability indicated by internal consistency, with Cronbach’s alpha coefficient of 0.73.” 

2. Result: Regression analysis only included variables that had significant association to PPD

Our response:

We thank Reviewer #2 for this comment. Only predictors that produced a p-value less than 0.05 in the chi-square tests were included for the regression analyses, as mentioned in line 148. We have re-analysed our data, and the predictors that fulfil the criteria included participants’ age, PoDLis score, ethnicity, monthly household income, and maternal employment status. PoDLis score & participants’ ethnicity have been taken out from the logistic regression analyses. Table 3 (page 12) has been corrected to reflect these changes. 

Reviewer #3: 

1. The manuscript is well written and logical flow. The research methods used is appropriate. However, the knowledge gained from this research is not new in the filed of PDD. The selected factors are so simple and are already identified from the literature.

Our response:

We thank Reviewer #3 for their comments. The contributions of our study include an update on the latest incidence of PPD in Malaysia’s capital, Kuala Lumpur, which was last reported 20 years ago by Grace et al. (2001) and Koo et al. (2003). Our study is also the first to report and explore the association between socioeconomic factors and PPD literacy with PPD incidence PPD knowledge and awareness among Malaysian mothers. We believe our findings are important as PPD has specific cultural implications and adds to the effort for targeted public education campaigns that could be more efficient for those who are more likely to develop PPD symptoms during antenatal and early postpartum periods.

End of response.

---

## [Editor Report · Decision Letter 1]

27 Oct 2021

Incidence and predictors of postpartum depression among postpartum mothers in Kuala Lumpur, Malaysia: a cross-sectional study

PONE-D-21-08937R1

Dear Dr. Hairol,

We’re pleased to inform you that your manuscript has been judged scientifically suitable for publication and will be formally accepted for publication once it meets all outstanding technical requirements.

Kind regards,

Hanani Tabana, Ph.D

Academic Editor

PLOS ONE
---

## [Editor Report · Acceptance letter]

29 Oct 2021

PONE-D-21-08937R1 

Incidence and predictors of postpartum depression among postpartum mothers in Kuala Lumpur, Malaysia: a cross-sectional study 

Dear Dr. Hairol:

I'm pleased to inform you that your manuscript has been deemed suitable for publication in PLOS ONE. Congratulations! Your manuscript is now with our production department. 

Kind regards, 

on behalf of

Dr. Hanani Tabana 

Academic Editor

PLOS ONE